# Osthole Prevents Heart Damage Induced by Diet-Induced Metabolic Syndrome: Role of Fructokinase (KHK)

**DOI:** 10.3390/antiox12051023

**Published:** 2023-04-28

**Authors:** Fernando E. García-Arroyo, Guillermo Gonzaga-Sánchez, Alejandro Silva-Palacios, Francisco Javier Roldán, María L. Loredo-Mendoza, Yamnia Quetzal Alvarez-Alvarez, Jesus A. de los Santos Coyotl, Kevin A. Vélez Orozco, Edilia Tapia, Horacio Osorio-Alonso, Abraham S. Arellano-Buendía, José L. Sánchez-Gloria, Miguel A. Lanaspa, Richard J. Johnson, Laura Gabriela Sánchez-Lozada

**Affiliations:** 1Department of Cardio-Renal Physiopathology, Instituto Nacional de Cardiología Ignacio Chávez, Mexico City 14080, Mexico; 2Department of Cardiovascular Biomedicine, Instituto Nacional de Cardiología Ignacio Chávez, Mexico City 14080, Mexico; 3Department of External Consultation, Instituto Nacional de Cardiología Ignacio Chávez, Mexico City 14080, Mexico; 4Department of Pathology, Instituto Nacional de Cardiología Ignacio Chávez, Mexico City 14080, Mexico; 5Renal Diseases and Hypertension, University of Colorado, Aurora, CO 80045, USA

**Keywords:** cardiac hypertrophy, polyol pathway, hyperuricemia

## Abstract

There is increasing evidence that either ingested or produced fructose may have a role in metabolic syndrome. While not commonly considered a criterion for metabolic syndrome, cardiac hypertrophy is often associated with metabolic syndrome, and its presence carries increased cardiovascular risk. Recently it has been shown that fructose and fructokinase C (KHK) can be induced in cardiac tissue. Here we tested whether diet-induced metabolic syndrome causes heart disease associated with increased fructose content and metabolism and whether it can be prevented with a fructokinase inhibitor (osthole). Male Wistar rats were provided a control diet (C) or high fat/sugar diet for 30 days (MS), with half of the latter group receiving osthol (MS+OT, 40 mg/kg/d). The Western diet increased fructose, uric acid, and triglyceride concentrations in cardiac tissue associated with cardiac hypertrophy, local hypoxia, oxidative stress, and increased activity and expression of KHK in cardiac tissue. Osthole reversed these effects. We conclude that the cardiac changes in metabolic syndrome involve increased fructose content and its metabolism and that blocking fructokinase can provide cardiac benefit through the inhibition of KHK with modulation of hypoxia, oxidative stress, hypertrophy, and fibrosis.

## 1. Introduction

Metabolic syndrome (MS) (hypertriglyceridemia, abdominal obesity, impaired glucose tolerance, insulin resistance, decreased HDL cholesterol, and hypertension) affects nearly one-quarter of the adult population in both the USA and Mexico and is a major risk factor for cardiovascular disease (CVD) [1]. While multiple mechanisms likely drive metabolic syndrome, fructose, either ingested or produced endogenously, has been shown to confer risk [2]. Our group has shown that fructose is unique in its ability to induce metabolic syndrome, which is mediated by the specific metabolism of fructokinase C (KHK-C) that leads to local uric acid generation, mitochondrial oxidative stress, and impairment in ATP production [3].

While not precisely a criterion for metabolic syndrome, cardiac hypertrophy is commonly associated with and confers increased cardiovascular risk. Dietary fructose intake in rats can also induce cardiac metabolic stress [4] and early signs of diastolic dysfunction, mitochondrial alterations, apoptosis, and oxidative stress [4,5]. It has recently been shown that fructose production can be induced in cardiac tissue via the polyol pathway and that with local hypoxia, there can also be the induction of KHK-C [4,5,6,7]. One study also found that hypoxia-mediated cardiac hypertrophy was induced by local fructose metabolism as determined using the KHK knockout mouse [6]. 

These findings suggest that the cardiac changes that occur with metabolic syndrome might also represent the direct effects of dietary and/or local fructose metabolism. To test this hypothesis, we provided rats with a Western or control diet for 30 days. Then we investigated if local fructose production and metabolism occurred in the heart and if it was associated with the development of cardiac hypertrophy. We also included a group that was administered an inhibitor of KHK-C that we have previously shown can block other metabolic effects of fructose [8].

## 2. Materials and Methods

### 2.1. Ethical Approval

This study was approved by the Internal Animal Care and Use Committee of the Instituto Nacional de Cardiología Ignacio Chávez (INC/CICUAL/013/2019) and conducted under the current Guide of Care and Use of Laboratory Animals, published by the Mexican Federal Regulation for Animal Experimentation and Care (NOM-062-ZOO-2001) and the National Institutes of Health.

### 2.2. Experimental Protocol

Three groups of male Wistar rats of 220–240 g of body weight (*n* = 7/group) were kept in individual acrylic cages for 1 week for acclimation with a black holder, a wood chip, and nesting and bedding paper material (Green soft, RGS, Mexico City, Mexico). The groups were followed for 30 days: control (C), healthy animals feeding with regular diet and tap water, metabolic syndrome (MS), feeding with a Western-type diet (high fat/sugar diet) and a sweetened beverage (3.8% glucose, 7.2% fructose) [8,9], metabolic syndrome + osthole (MS+OT), osthole was administered at a dose of 40 mg/kg in the food. 

Our work group previously reported this model and diet composition [8]. At the end of the study, animals were anesthetized with 3% isofluorane (Somno Suite, Kent Scientific Corporation, Torrington, CT, USA) and exsanguinated by abdominal aorta punction. Plasma and urine samples were collected and frozen at −70 °C. For further analysis, tissue (kidney and heart) was dissected and frozen in liquid nitrogen.

### 2.3. Body Weight and Feeding

Food was offered in powder with water (1 g/mL) as a paste. The osthole dose was mixed with the food and modified daily according to body weight. The amount of food was controlled and changed in all groups weekly (week 1: 25 g; week 2: 28 g; week 3: 31 g; week 4: 34 g). The consumption of fluids, food, and body weight was quantified and registered daily.

### 2.4. Systolic Blood Pressure (SBP)

SBP was assessed with a tail-cuff sphygmomanometer (IN125/R/ ADI Instruments, Colorado Spring, CO, USA) in conscious, trained rats placed in an incubator at 40 °C for 5 min to dilate the tail vein. The mean of 3 measurements was reported. 

#### Evaluation of Cardiac Function by Echocardiography

Rats were anesthetized with a low dose of sodium pentobarbital (19 mg/100 g body weight, IP) for echocardiographic analysis using a Sonos 550 echocardiographer (Koninlijke Phillips Electronics, Eindhoven, The Netherlands) with a 12 MHz transducer. Briefly, the parasternal short and long axes were analyzed by two-dimensional M-mode echocardiography in each animal. Left ventricle (LV) cavity and thickness were measured to calculate ejection fraction (EF) using the formula %EF = Y + [(100 − Y) × 0.15)], where Y = [LVEDd2 − LVEDs2/LVEDd2] × 100, as well as fractional shortening (FS) by the formula %FS = [(LVEDd − LVEDs/LVEDd) × 100], where LVEDd and LVEDs are the LV dimension at end-diastole and end-systole, respectively, according to previous reports [10,11]. At the end of the evaluation, the animals were allowed to recover for a few days before being euthanized.

### 2.5. Systemic Parameters

To determine the induction of metabolic syndrome, we measured different parameters. Quantification of triglycerides, HDL, and uric acid was performed using commercial kits (Sekisui Diagnostics, Burlington, MA, USA). Glucose was measured with a commercial kit according to the manufacturer’s instructions (Glucose-LQ, Spinreact, Girona, Spain). All parameters were read in a hybrid multimode reader (Sinergy H1, Biotek-Agilent, Santa Clara, CA, USA).

### 2.6. Heart Fructokinase (KHK) Activity Assessment

The activity of KHK was measured indirectly through ATP consumption, as previously reported [8]. Fifty micrograms of protein were incubated in buffer imidazole 50 mM and KAc 1 M. Samples were exposed to 5 mM fructose and 1 mM ATP for 2 h. ATP concentrations were measured by luminescence in a hybrid multimode reader (Sinergy H1, Biotek-Agilent, Santa Clara, CA, USA). Results were corrected per milligram of protein and expressed in relative luminescence units (RLU).

### 2.7. Heart Fructose and Uric Acid Content

Cardiac fructose was extracted from 50 mg of left ventricle tissue with perchloric acid and measured using the anthrone method [8,12]. Cardiac uric acid was extracted from 50 mg of tissue, homogenized in extraction buffer (25 mM HEPES, 10 mM KCl, 1 mM DTT, 1 mM EDTA, and 5%NP-40 substitute), and three episodes of cold-heat shocks (liquid nitrogen-thermo block) [13] and measured with a commercial kit accordingly to the manufacturer instructions (Sekisui Diagnostics, Burlington, MA, USA). Both parameters were read in a hybrid multimode reader (Sinergy H1, Biotek-Agilent, Santa Clara, CA, USA). Results were corrected per milligram of protein.

### 2.8. Western Blot Analysis

Protein expression was evaluated by western blot in left ventricle tissue homogenized in a RIPA buffer [14]. Thirty µg of protein were loaded in SDS-Polyacrylamide Gel Electrophoresis (Mini Protean, Bio-Rad, Hercules, CA, USA). Proteins were transferred to a nitrocellulose membrane (Criterion Blotter, Bio-Rad, Hercules, CA, USA) and blocked with non-fat dry milk 5% (Bio-Rad, Hercules, CA, USA) in a TBS-Tween buffer for 1 h. All antibodies used were incubated overnight at 4 °C. Antibodies used were as follows: Fructokinase (Genetex, GTX109591, 1:5000 dilution, Irvine, CA, USA), hypoxia-inducible factor 1α (Genetex, GTX127309, 1:7500 dilution), Cardiac troponin 1 (Genetex, GTX134489, 1:2000 dilution), Brain natriuretic peptide (Abcam, ab19645, 1:5000 dilution), Atrial natriuretic peptide (Abcam, ab225844 dilution, Cambridge, UK), CD63 (Genetex, GTX132953, 1:2000 dilution), CD9 (Genetex, GTX66709, 1:1000 dilution), CD81 (Genetex, GTX31381, 1:2000 dilution), Transforming growth factor beta (Genetex, GTX45121, 1:5000 dilution), alpha-smooth muscle actin (Genetex, GTX100034) 1:3000 dilution), Histone H3 (Genetex, GTX122148, 1:10,000 dilution), and beta-actin (Genetex, GTX109639, 1:10,000 dilution). The secondary antibody was incubated for two hours at room temperature (Antirabit-IgG, HRP-linked antibody, Cell Signaling, 7074S, Danvers, MA, USA). The chemiluminescence was determined with a commercial kit (SuperSignal West Pico PLUS Chemiluminescent Substrate, Thermo Scientific, Waltham, MA, USA). 

### 2.9. Oxidative Stress Markers

We assessed protein carbonylation and lipid peroxidation in cardiac tissue using previously reported methods [8,15]. Cardiac left ventricle (50 mg) was homogenized in PBS 20 mM, washed with streptomycin sulfate, and incubated with 2-4-dinitrophenylhydrazine (DNPH); the interaction protein–carbonyl group was estimated using the reaction with DNPH. Lipid peroxidation was measured by the 4-hydroxinonenal presence in the left cardiac ventricle using a standard curve of 1,1,3,3 tetramethoxypropane [8,15]. Total glutathione concentration was measured by an enzymatic method in left cardiac ventricle homogenates (50 mg) by a method previously reported [16]. The three markers were read in a hybrid multimode reader (Sinergy H1, Biotek-Agilent, Santa Clara, CA, USA). Results were corrected per milligram of protein.

### 2.10. Nrf2 Factor Transcription Assay by Trans-Activation

The binding of transcription factor NF-E2 p45-related factor 2 (Nrf2) with the antioxidant response element (ARE) was assessed with an ELISA assay kit (Nrf2 Transcription Factor Assay Kit, Colorimetric, Abcam). Five micrograms of left ventricle protein was incubated with oligonucleotide ARE consensus binding site (5′ GTCACAGTGACTCAGCAGAATCTG–3′). Detection was performed with specific antibodies and measured by colorimetric (HRP catalyzed reaction) at 450 nm in a hybrid multimode reader (Sinergy H1, Biotek-Agilent, Santa Clara, CA, USA).

### 2.11. Plasma Extracellular Vesicles Isolation

Plasma extracellular vesicles (EVS) were isolated with a commercial kit according to manufacturer instructions (Total Exosome Isolation Kit for plasma, Invitrogen-Thermo Scientific, Waltham, MA, USA) (Appendix A). For posterior analysis, the final supernatant and pellet (EVS) fractions were frozen at −20 °C. For western blotting determinations, extracellular vesicles were resuspended and lysed in a RIPA buffer. Characterization of exosomes was also assessed by western blotting.

### 2.12. Histological Analysis

The hearts were harvested and immersed in 10% neutral phosphate-buffered saline (PBS) formalin. After a 24 h fixation period, they were weighed and trimmed before sectioning at the mid-ventricular region, obtaining one LV base ring and one apical ring from each heart. Fixed rings were processed as usual and embedded in paraffin, then 5 µm thick sections were obtained, mounted onto glass slides, and stained with hematoxylin and eosin (HE).

HE stained sections were evaluated with a Zeiss Axioplan 2 light microscope (Zeiss, Jena, Germany). Low magnification fields throughout the entire myocardial area of one basal ring and one apical ring per heart were used to quantify injured myocardial areas. Myocardial injured areas and the total area of LV myocardium were traced manually in digital images, measured automatically by the microscope camera software (Axiovision Rel 4.8.2 Zeiss), and expressed as a percentage. A 400× magnification was used to characterize myocyte injury and inflammatory infiltration.

### 2.13. Immunohistochemistry

For immunohistochemistry, 5 µm thick basal and apical heart ring sections were deparaffinized and boiled in citrate buffer pH 6.5 for 20 min for antigen retrieval. Then the slides were incubated overnight at 4 °C with the primary antibody (Anti-Fructokinase, Genetex, GTX109591) at 1/50 dilution, washed with PBS-Tween, and incubated for 60 min at room temperature with anti-rabbit biotinylated secondary antibody (Invitrogen, Cat. 65-6140)), washed with the same solution. Finally, the sections were incubated 30 min in HRP-conjugated streptavidin for 30 min, and DAB was utilized as the chromogen. 

Five fields of each sample were analyzed to quantify endothelial capillary cells positive for anti-KHK using an 80-point ocular grid with 200× magnification. The results were expressed as a percentage.

### 2.14. Statistical Analysis

Data were analyzed using Graph Pad Prism 7.1 (San Diego, CA, USA), and results were presented as the mean ± standard deviation and analyzed by one-way ANOVA. Statistical differences were established as * *p* ≤ 0.05, ** *p* ≤ 0.01, *** *p* ≤ 0.001, and **** *p* ≤ 0.0001. Post hoc analysis was performed using Tukey’s multiple comparison test.

## 3. Results

### 3.1. Body Weight and Fluid and Food Consumption

At the end of the study, we observed that MS and MS+OT groups gained more weight than the C group. Interestingly, the MS+OT group significantly gained less weight than the MS group, although the mean food and fluid intake was similar. Therefore, the osthole partially reduced the increase in body weight (Figure 1).

### 3.2. Markers of Metabolic Syndrome

The rats receiving the Western diet developed elevated plasma triglycerides, glucose, and uric acid compared to control rats and demonstrated higher systolic blood pressure. Osthole treatment partially but significantly reduced these metabolic manifestations (Figure 2). These data agree with our previous studies [8].

The expression of KHK-C was almost undetectable in the heart tissue of control rats but was markedly increased in rats receiving the Western diet. Rats on the Western diet that received osthol showed significantly less upregulation of KHK-C. A consequence of fructose metabolism is intracellular ATP depletion. Osthole administration significantly blocked this reduction in ATP (Figure 3A,B). 

Diet and endogenous production regulate cardiac fructose content [2,17]. We found that in control rats, heart fructose was undetectable; however, the Western diet significantly increased heart fructose concentration, and this effect was mildly but significantly decreased by osthole. Accordingly, the Western diet significantly increased the byproduct of fructose metabolism, uric acid, and again osthole mildly but significantly prevented such effect. Induction of KHK-C in the heart has been previously reported to be mediated by the activation of the hypoxia-inducible factor 1 alpha (HIF-1α) [7]. Consistently with this prior work, we found that HIF-1α was overexpressed in metabolic syndrome rats, compared to control rats, and that osthole administration could partially prevent these findings (Figure 3B).

### 3.3. Evaluation of Cardiac Function by Echocardiography and Systemic Markers of Cardiac Damage

Cardiac weight was increased in the rats with metabolic syndrome, and the lung/body weight ratio and heart weight/body weight ratio were also increased, and these changes were partially blocked by treatment with osthole. On the other hand, the echocardiographic data did not show a thickening of the ventricular wall or an increase in chamber dilation, indicating that at this time, MS did not induce cardiac dysfunction measured by this parameter. (Figure 4, Table 1).

### 3.4. Cardiac Oxidative Stress Assessment and Nrf2 Activation

Oxidative stress was assessed by measuring the concentrations of lipid peroxidation (measured by 4-HNE) and protein carbonylation (measured by complex DNPH-protein). Both oxidative stress markers increased significantly in the MS group vs. C group, and osthole treatment partially prevented oxidative stress in cardiac tissue (Figure 5). On the other hand, the total concentration of glutathione diminished in the MS group, while osthole treatment increased this parameter significantly. Plasma lipid peroxidation measured by malondialdehyde (MDA) is a reliable marker of oxidative stress burden and cardiovascular risk [18]. MDA concentrations increased markedly at 15 days in both the MS and MS+OT groups without significant differences between groups. At 30 days, osthole-treated rats showed a mild reduction in MDA concentration compared to Western diet-induced MS rats. Additionally, MS induced a significant reduction of Nrf2-ARE binding activation, which was prevented by an osthole in the left cardiac ventricle (Figure 5).

### 3.5. Fibrosis Assessment in Cardiac Tissue and Extracellular Vesicles

We evaluated the expression of α-smooth muscle actin (α-SMA) and transforming growth factor beta 1 (TGFβ-1) in the cardiac left ventricle by western blot. At the end of the follow-up, we found that MS induced a significant increment in α-SMA and TGFβ-1. Osthole treatment did not prevent α-SMA overexpression and slightly but significantly reduced the expression of TGFβ-1 (Figure 6) in cardiac tissue. 

On the other hand, the expression of TGFβ-I in extracellular vesicles isolated from plasma increased similarly after 15 days in MS and MS+OT, but at 30 days, osthole treatment significantly reduced their expression (Figure 6).

### 3.6. Histological Analysis

The administration of the Western diet resulted in histologic evidence for myocardial injury, with myocytes demonstrating “an empty look appearance”, often surrounded by a mild to moderate mononuclear inflammatory infiltrate. Focal areas showed fibrosis and a decrease in myofibrils. Rats on the control diet showed no abnormal histology, while injury was reduced in rats on the Western diet that received osthole, with a lower percentage of myocardial injured areas found (Figure 7).

### 3.7. KHK Immunohistochemistry

MS induced a significant positivity for KHK by immunohistochemistry in endothelial cells of capillaries and a few cardiomyocytes compared to the control group. Osthole treatment prevented KHK overexpression in heart tissue (Figure 8).

## 4. Discussion

Cardiovascular disease remains a significant health problem worldwide. Nevertheless, the etiology is multifactorial, and studying the different causal factors is urgently needed. Metabolic syndrome is highly prevalent in children and adults and is treatable and preventable; hence the importance of investigating its pathological consequences in different target organs. Our study showed that de novo heart fructose metabolism might play an essential role in developing cardiac damage induced by MS. Thus, the activation of KHK and HIF-1α are likely critical factors contributing to mediating heart disease in this condition. 

We and others have previously shown that the high sugar-high fat Western diet induced metabolic syndrome in rodents and humans [8,9,19,20,21]. Additionally, we previously showed that osthole treatment reduced the manifestation of metabolic syndrome with concomitant protection of renal damage [8]. Various protective mechanisms have been attributed to osthole, including the ability to modulate lipogenesis and oxidative stress [8,22]. In addition, osthole is an inhibitor of KHK and xanthine oxidase, thus reducing uric acid concentrations [8].

In agreement with our findings, we showed MS-induced mild cardiac alterations after four weeks of diet exposure. In this study, we report that the early stimulation of KHK activity and HIF1α and aldose reductase (Appendix A) overexpression at the heart level may be part of the cardiac hypertrophic growth process. We also found a significant increase in cardiac fructose and uric acid concentrations. In this regard, there is evidence of a significant increase in cardiac fructose content and metabolism in the heart alterations induced by diabetes [2]. By immunohistochemistry analysis, we observed an increased expression of KHK protein in the myocardial capillary endothelial cells of MS rats, notably diminished by osthole.

Interestingly, osthole also partially reduced the overexpression of HIF1α, and this effect has been previously reported in hypoxic colon cancer cells [23]. Therefore cardiac hypoxia and increased systemic fructose load likely stimulated fructose metabolism in the heart, as was previously described in other models of cardiac hypertrophy and diabetes [2,6]. Hypertension, cardiac hypertrophy, and systemic markers of the cardiac lesion and fibrosis were present in our MS model, but cardiac functional alterations by echocardiography were not observed. Nevertheless, the increased lung weight and LW/BW ratio observed in MS rats suggest a dysfunctional LV, and osthole was able to reduce those cardiac changes significantly.

Oxidative stress is essential in driving MS and cardiac damage [8,24,25]. We observed that osthole treatment reduced cardiac 4-HNE and protein oxidation and increased glutathione concentration. Antioxidant effects of osthole have been previously shown, including the activation of transcription factor Nrf2 that led to activating the expression of antioxidant enzymes (superoxide dismutase, catalase, glutamate-cysteine ligase, and glutathione peroxidase) [8,26]. Nrf2 can also modulate lipid and carbohydrate metabolism and participate in detoxification [27]. Those additional osthole-mediated effects likely also provided a salutary effect on cardiac alterations induced by MS.

A limitation of this study is that osthole was given in the diet of animals from the beginning of the study and not in rats with established MS. Another limitation is that the Western diet per se induced the activation of the polyol pathway, which could be generating endogenous fructose. However, the osthole partially prevented the overactivation of the polyol pathway and the KHK-C induced by MS in the heart. Thus, our studies disclosed the importance of fructose metabolism in MS-induced cardiac damage. We recommend further investigating this mechanism and implementing clinical trials using fructokinase inhibitors, as they have been proven safe in humans [28].

## 5. Conclusions

Osthole is a nutraceutical that ameliorates metabolic syndrome and cardiac alterations induced by a high sugar/high fat Western diet. Osthole reduced hypoxia, cardiac damage, oxidative stress, fibrosis, and activated Nrf2. Those effects were partially mediated through the blockade of KHK-mediated fructose metabolism at the cardiac level. The current feeding habits, which include an increased intake of high glycemic index/load and fructose, have the potential to induce and aggravate cardiac hypertrophy [29]. Particularly, fructose metabolism in the heart sustains hypertrophic growth [7]. In human studies, we believe it is worth testing osthole as a coadjuvant to treat MS and cardiac damage.

## Figures and Tables

**Figure 1 antioxidants-12-01023-f001:**
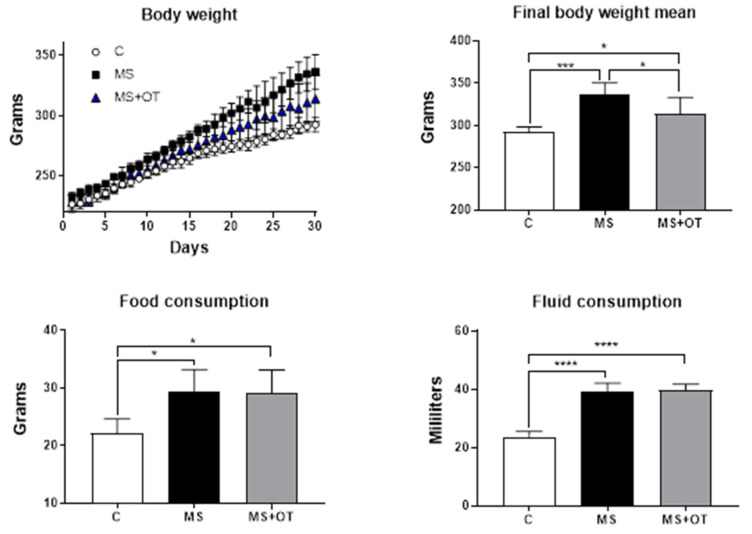
Effect of Western-type diet and osthole administration on body weight, fluid intake, and food consumption after 30 days follow-up. Data are presented as the mean ± standard deviation and were analyzed by one-way ANOVA. Analysis post hoc was performed using Tukey’s multiple comparison test. Statistical differences were established as * *p* ≤ 0.05, *** *p* ≤ 0.001, and **** *p* ≤ 0.0001.

**Figure 2 antioxidants-12-01023-f002:**
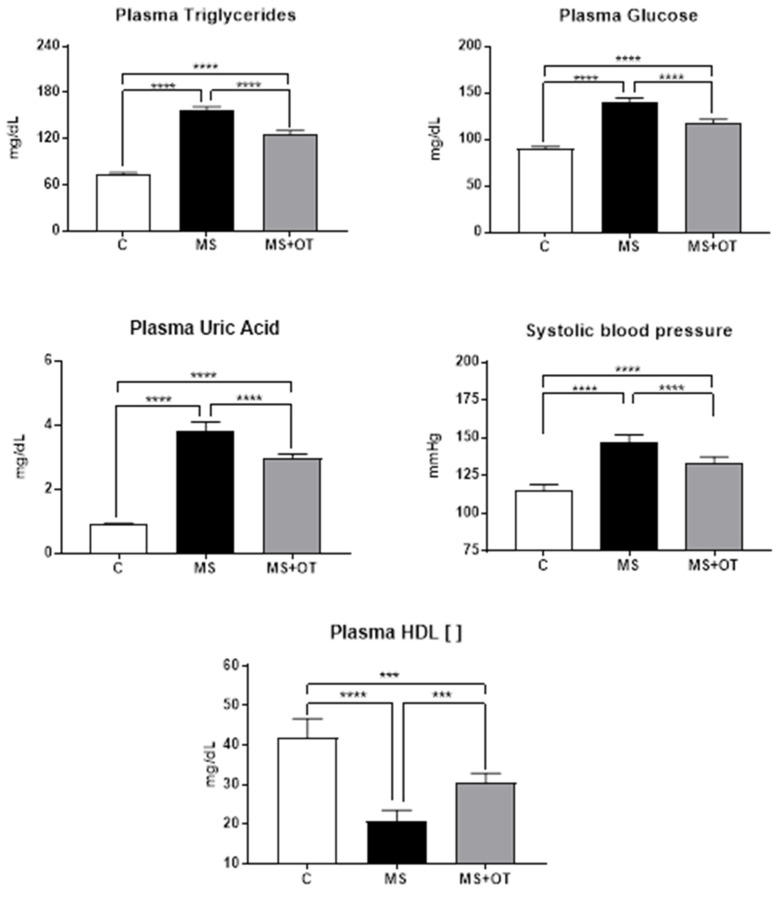
Effect of Paigen-type diet and osthole administration on plasmatic triglycerides, glucose, uric acid, HDL, and systolic blood pressure after 30 days follow-up. Data are presented as the mean ± standard deviation and were analyzed by one-way ANOVA. Analysis post hoc was performed using Tukey’s multiple comparison test. Statistical differences were established as *** *p* ≤ 0.001 and **** *p* ≤ 0.0001.

**Figure 3 antioxidants-12-01023-f003:**
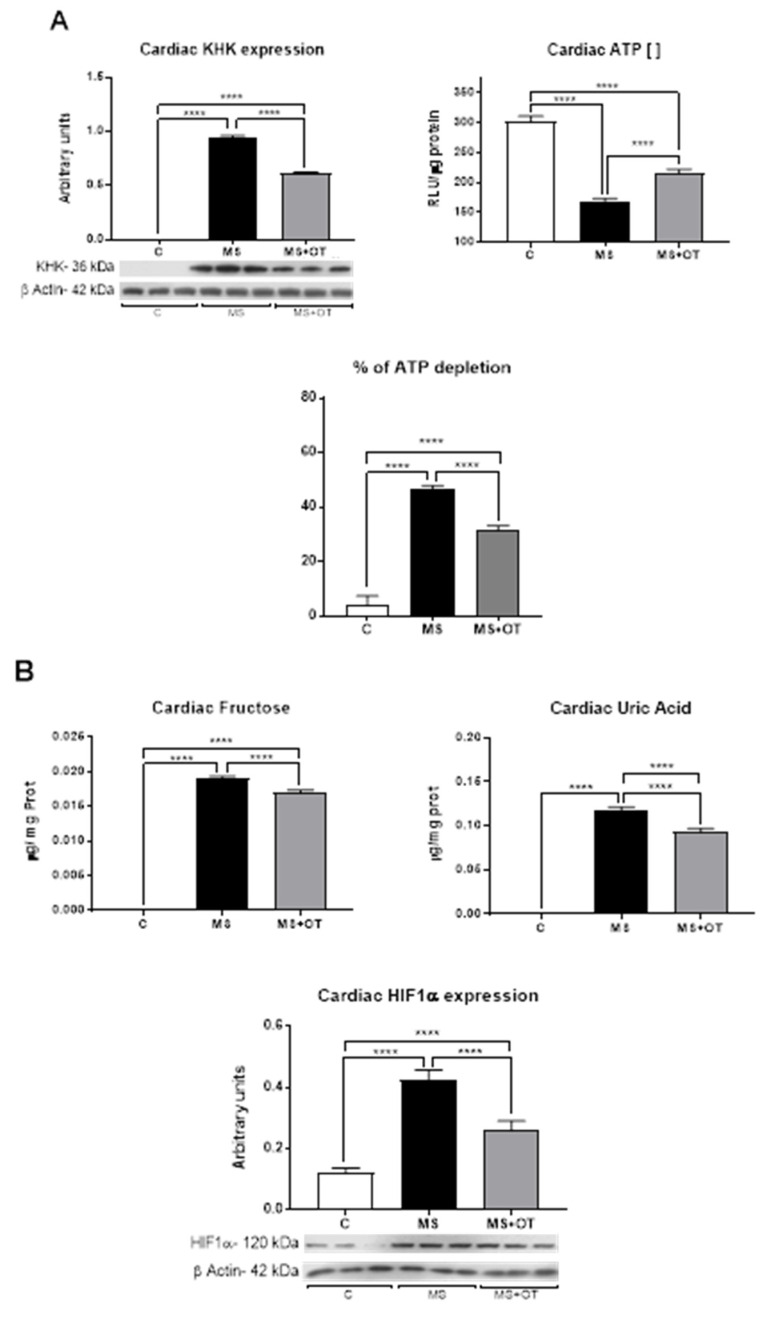
Effect of Paigen-type diet and osthole administration on cardiac KHK expression, activity (**A**), hypoxia-inducible factor 1 alpha (HIF-1α), and products of fructose metabolism (**B**) after 30 days follow-up. Data are presented as the mean ± standard deviation and were analyzed by one-way ANOVA. Analysis post hoc was performed using Tukey’s multiple comparison test. Statistical differences were established as **** *p* ≤ 0.0001.

**Figure 4 antioxidants-12-01023-f004:**
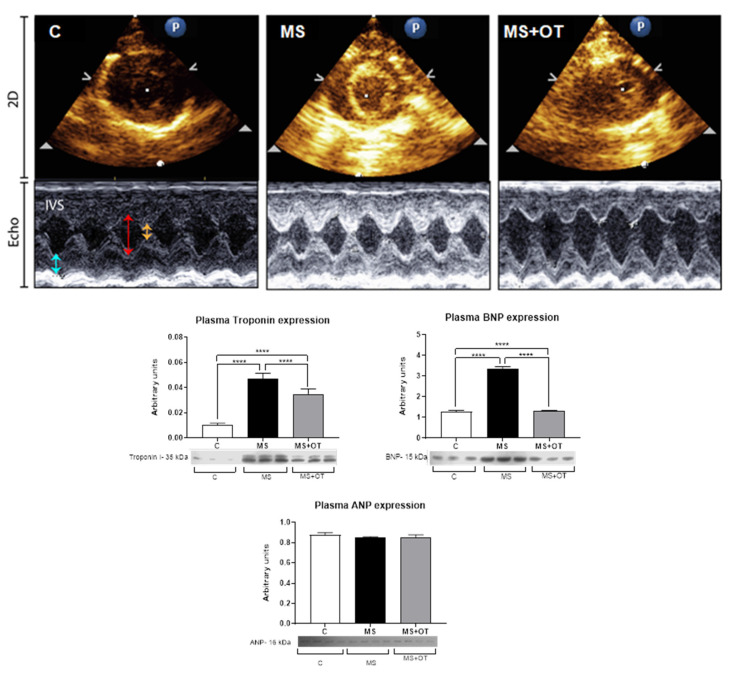
(**Upper panel**). Representative echocardiography (Echo) and 2D images. IVS: interventricular septum; LVEDd: LV dimension at the end of diastole (red arrow); LVEDs: LV dimension at the end of systole (yellow arrow); LVPW: LV dimension posterior wall (arrow blue). (**Lower panel**). Effect of Paigen-type diet and osthole administration on plasma cardiac Troponin I, BNP, and ANP after 30 days follow-up. The expression of troponin I, BNP, and ANP was corrected by total plasma protein concentrations. Data are presented as the mean ± standard deviation and were analyzed by one-way ANOVA. Analysis post hoc was performed using Tukey’s multiple comparison test. Statistical differences were established as **** *p* ≤ 0.0001.

**Figure 5 antioxidants-12-01023-f005:**
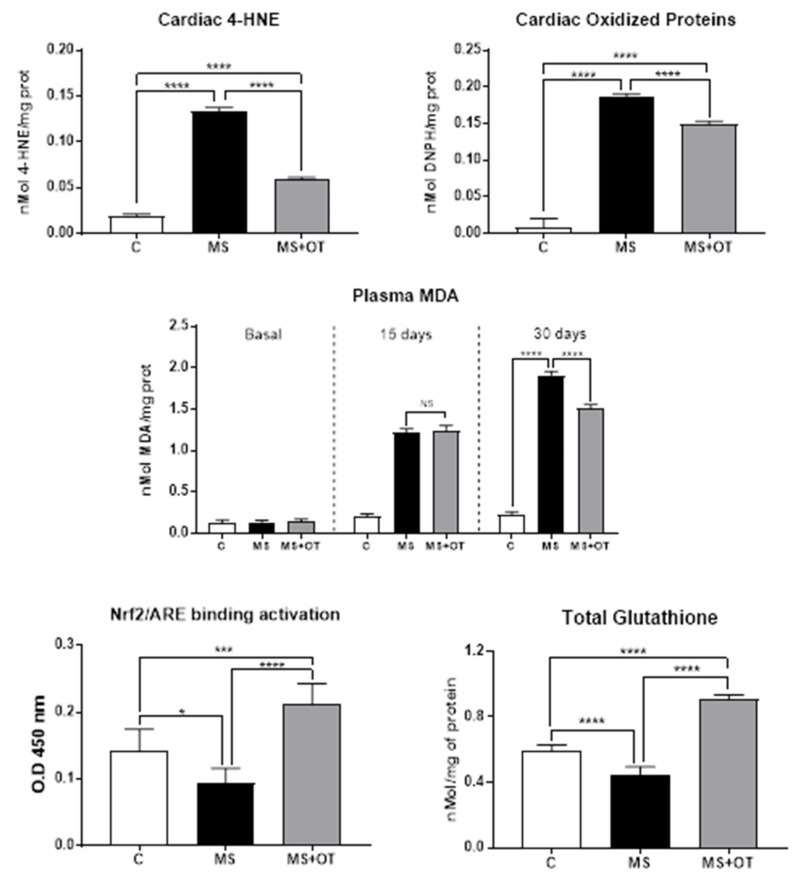
Effect of Paigen-type diet and osthole administration on oxidative stress markers after 30 days follow-up. Data are presented as the mean ± standard deviation and were analyzed by one-way ANOVA. Analysis post hoc was performed using Tukey’s multiple comparison test. Statistical differences were established as * *p* ≤ 0.05, *** *p* ≤ 0.001, and **** *p* ≤ 0.0001.

**Figure 6 antioxidants-12-01023-f006:**
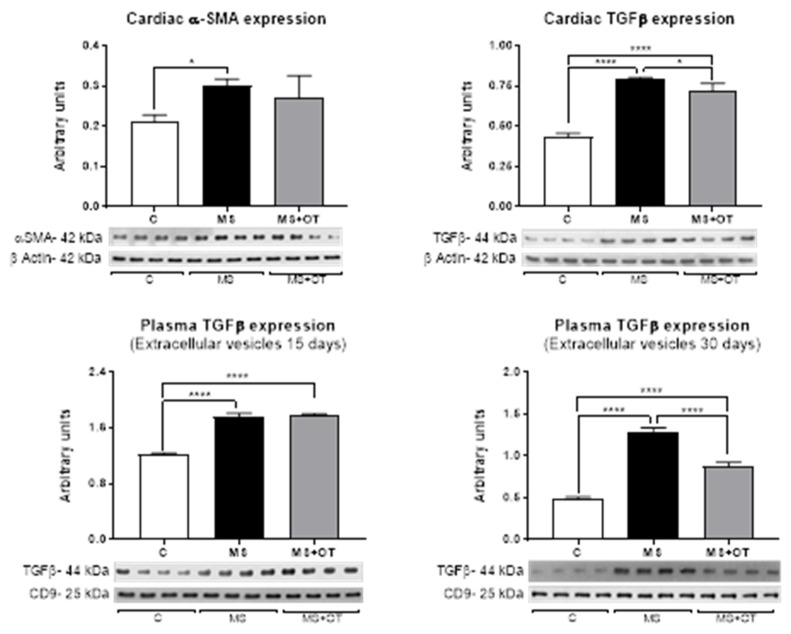
Effect of Paigen-type diet and osthole administration on cardiac fibrosis markers and TGFβ in EVS after 15 and 30 days follow-up. Data are presented as the mean ± standard deviation and were analyzed by one-way ANOVA. Analysis post hoc was performed using Tukey’s multiple comparison test. Statistical differences were established as * *p* ≤ 0.05 and **** *p* ≤ 0.0001.

**Figure 7 antioxidants-12-01023-f007:**
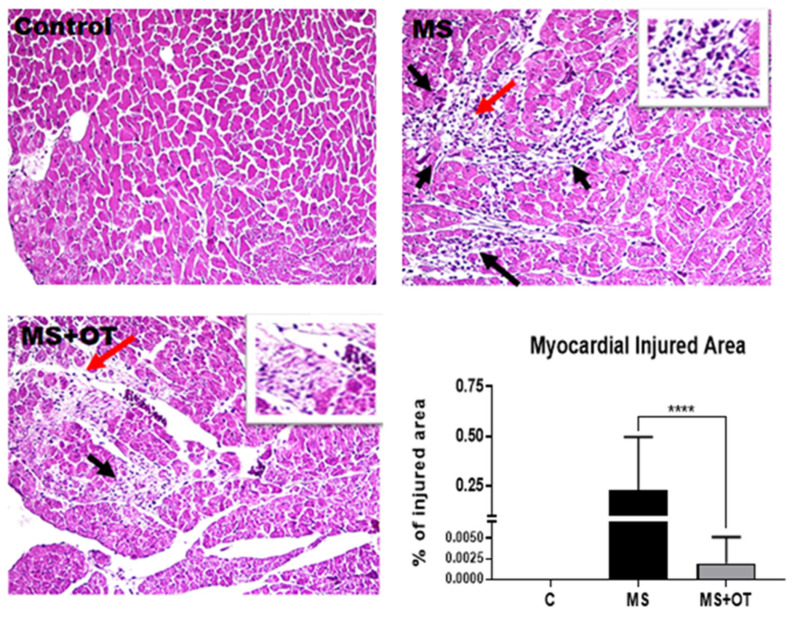
Effect of Paigen-type diet and osthole administration in cardiac inflammatory infiltrate in histological analysis after 30 days follow-up. Black arrows show myocytes with an empty appearance, surrounded by mononuclear inflammatory cells, especially macrophages. The red arrow points to a magnified area on the insert; 400× magnification. Data are presented as the mean ± standard deviation and were analyzed by one-way ANOVA. Analysis post hoc was performed using Tukey’s multiple comparison test. Statistical differences were established as **** *p* ≤ 0.0001.

**Figure 8 antioxidants-12-01023-f008:**
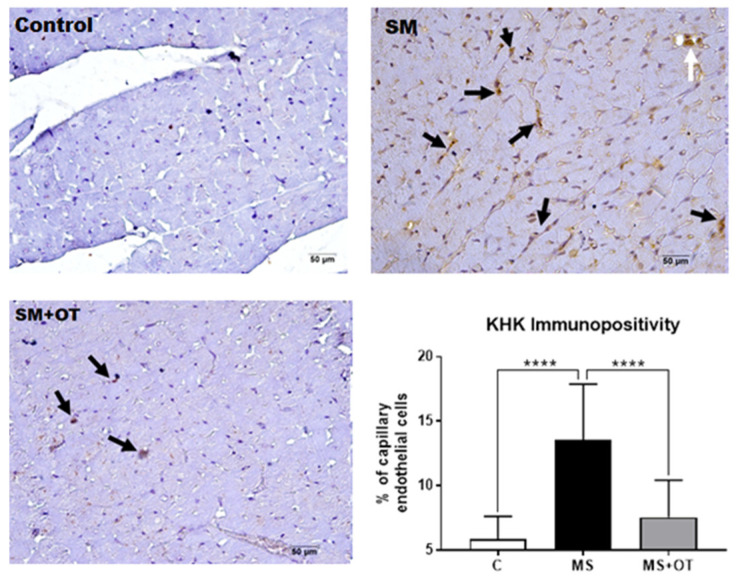
Effect of Paigen-type diet and osthole administration in cardiac KHK expression measured by immunohistochemistry after 30 days follow-up. Black arrows point to cardiac endothelial cells positive for KHK. White arrows represent three endothelial cells in a venule. Data are presented as the mean ± standard deviation and were analyzed by one-way ANOVA. Analysis post hoc was performed using Tukey’s multiple comparison test. Statistical differences were established as **** *p* ≤ 0.0001.

**Table 1 antioxidants-12-01023-t001:** Effect of Paigen-type diet and osthole administration on echocardiographic parameters after 30 days follow-up.

Parameter/ Group	C	MS	MS+OT
IVS (mm)	0.18 ± 0.019	0.205 ± 0.026	0.19 ± 0.012
LVEDd (mm)	4.96 ± 0.44	4.88 ± 0.39	4.62 ± 0.57
LVEDs (mm)	2.33 ± 0.53	2.06 ± 0.37	2.02 ± 0.43
LVPW (mm)	0.17 ± 0.019	0.19 ± 0.019	0.18 ± 0.016
EF (%)	81.4 ± 5.62	84.4 ± 6.1	83.3 ± 6.5
FS (%)	53.4 ± 7.36	57.7 ± 7.3	56.3 ± 8.5
HR (bpm)	424.5 ± 34.72	378.9 ± 51.4	412.5 ± 32.1
Body weight (g)	292.5 ± 5.82	336.4 ± 14.30	314 ± 15.9
Heart weight (HW, g)	1.05 ± 0.07	1.37 ± 0.05 ^a^	1.26 ± 0.03 ^b^
Lung weight (LW, g)	1.90 ± 0.13	3.83 ± 0.39 ^a^	2.91 ± 0.20 ^b^
Tibial length (TL, cm)	3.92 ± 0.14	4.20 ± 0.20	3.84 ± 0.83
HW/TL (g/cm)	0.27 ± 0.02	0.33 ± 0.02	0.35 ± 0.11
LW/TL (g/cm)	0.49 ± 0.04	0.91 ± 0.10 ^a^	0.82 ± 0.32 ^a^
HW/BW (g/kg)	3.52 ± 0.20	4.56 ± 0.24 ^a^	4.38 ± 0.29 ^a^
LW/BW (g/kg)	6.38 ± 0.55	12.72 ± 1.38 ^a^	10.16 ± 1.26 ^a,b^
Functional parameters	(*n* = 7)	(*n* = 8)	(*n* = 7)

Data are the mean ± SD of at least 6 different animals in each experimental group. ^a^
*p* ≤ 0.05 vs. C; ^b^
*p* ≤ 0.05 vs. MS. I.V.S.: interventricular septum; LVEDd: LV dimension at the end of diastole; LVEDs: LV dimension at the end of systole; LVPW: LV dimension posterior wall, EF: ejection fraction; FS: fractional shortening; HR: heart rate; bpm; beats per minute.

## Data Availability

The data presented in this study are available in the article and Appendix A.

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
