# Peer review of "Osthole Prevents Heart Damage Induced by Diet-Induced Metabolic Syndrome: Role of Fructokinase (KHK)"

_antioxidants, 2023, doi:10.3390/antiox12051023_

Round 1

Reviewer 1 Report

Metabolic syndrome is a global epidemic and continuously increasing causing several disturbances including increased diabetes, increased CVD etc. There exist definitions for MS that typically include factors such as hyperTG, abdominal obesity, impaired glucose tolerance, insulin resistance, decreased HDL-C, and hypertension. The authors have studied here cardiac hypertrophy that previously has also been connected to MS. Their main aim was to investigate the role of fructose in cardiac tissue and its effect on cardiac derangements including hypertrophy, oxidative parameters, uric acid level, TG levels etc.. The authors raised the role of fructokinase C (KHK) here and its regulatory role was studies using also its inhibitor, Osthole. The used methods are relevant to test the various aims of the study and the manuscript is clearly written. There are several issues, however, that need further discussions.

MAJOR COMMENTS

1. The authors showed plasma TG elevation in their rat model after the Paigen/Paigen + sugar diet. What was the effect on HDL levels? 

2. Why did the authors use only male rats and not include both genders in their study?

3. Polyol (or sorbitol) pathway is a 2-step process converting glucose to fructose. This pathway is usually associated to reductions in nitric oxide (NO) and glutathione that further increases damage to cardiac tissue. What were the levels of NO and glutathione and did osthole increase their levels once connected to Paigen diet+sugars ?

4. What was effect of the Paigen diet +/- sugars on the cardiac tissue lipid balance and what was the effect of osthole on the cardiac lipids?

5. Did the Paigen diet + sugars elevate cardiac tissue reactive oxygen species, ROS? 

6. Global problem with elevated consumption of beverages and other food containing high levels of high fructose corn syrup (HFCS) is well known as well as the unhealthy consequences. The authors should add a paragraph in their Discussion on the translation of their data to human context and whether there exist reports on the cardiac hypertrophy associated with fructose consumption in humans.

MINOR COMMENTS

1. Line 40, add reference after the sentence "Our group..."

2. Fig. 7; add to the figure legend what do the red and black arrows represent?

3. Fig. 8; Identify in the figure legend what the black and white arrows represent?

Author Response

Please see attachement.

Reviewer 2 Report

In this study, the authors tested whether diet-induced metabolic syndrome would lead to heart disease associated with increased fructose content and metabolism, and whether it can be prevented with osthole. Overall, the work is well done. But there are also some problems. I summarize some questions and suggestions as follows:

1. The experimental design did not establish mice with osthole added separately, and the three groups designed in the article reflected that OT could reverse the damage of MS, but the beneficial effect of a single OT was not reflected. It is not known whether the corresponding experimental indexes will change inconsistently with expectations after the variable osthole is added to mice alone.

2. The main tissue of the experiment involved in this article is the left ventricle of mouse heart tissue, but there are only 7 mice in each of the three groups, and a large number of enzyme activity tests, as well as Western blot, were done, are the experiments done in biological replicates? How were the experimental samples assigned?

3. Is it accurate that in the two graphs of cardiac fructose as well as cardiac uric acid in Figure 3, the content of group C is zero? Would endogenous fructose not be produced either?

4. How to determine in 3.4 that the significant reduction in Nrf2-ARE binding activation was caused by Osthole blocking Nrf2-ARE binding activation in the left ventricle, and what is the interaction between Nrf2-ARE and Osthole?

5. What is the white arrow in Figure 8? And is the naming in the upper left corner of the picture a marking error?

6. What is the activation of the polyol pathway observed in line 356, what is meant here? And does this conflict with the cardiac fructose as well as cardiac uric acid content of 0 in group C in Figure 3?

7. In line 23 of the abstract, "triglycerides concentrations" is suggested to be changed to "triglyceride concentrations" or "the concentration of triglycerides".

8. In the text, "western diet" is suggested to be changed to "Western diet".

9. In the introduction, line 46, "there can also be induction of KHK-C" is suggested to be changed to "there can also be the/an induction of KHK-C ".

10. In the introduction, line 51, "we provided rats a western or control diet for 30 days." is suggested to be changed to "we provided rats with western or control diet for 30 days.".

11. Line 60 in 2.1published by the Mexican Federal Regulation for animal experimentation and care (NOM-062-ZOO-2001)and National Institutes of Health. is suggested to be changed to "published by the Mexican Federal Regulation for Animal Experimentation and Care (NOM-062-ZOO-2001)and the National Institutes of Health. ".

12. Line 101 in 2.5 "the manufacturer instructions" is suggested to be changed to "the manufacturer's instructions" or "instructions of the manufacturer".

13. Some of the problems of writing specification, such as the lack of space in line 197 (Figure1) in 3.1 and the wrong use of the definite article the.

14. In the text, "MS induced" is suggested to be changed to "MS-induced".

Author Response

Please see atachment

Round 2

Reviewer 1 Report

I have no further comments.